# The miR-27a/FOXJ3 Axis Dysregulates Mitochondrial Homeostasis in Colorectal Cancer Cells

**DOI:** 10.3390/cancers13194994

**Published:** 2021-10-05

**Authors:** Giovannina Barisciano, Manuela Leo, Livio Muccillo, Erica Pranzini, Matteo Parri, Vittorio Colantuoni, Maria Letizia Taddei, Lina Sabatino

**Affiliations:** 1Department of Sciences and Technologies, University of Sannio, Via Francesco de Sanctis, 82100 Benevento, Italy; barisciano@unisannio.it (G.B.); manleo@unisannio.it (M.L.); livio.muccillo@unisannio.it (L.M.); colantuoni@unisannio.it (V.C.); 2Department of Experimental and Clinical Biomedical Sciences, University of Florence, Viale Morgagni 50, 50134 Firenze, Italy; erica.pranzini@unifi.it (E.P.); matteo.parri@unifi.it (M.P.); 3Department of Experimental and Clinical Medicine, University of Florence, Viale Morgagni 50, 50134 Firenze, Italy; marialetizia.taddei@unifi.it

**Keywords:** colorectal cancer, FOXJ3, miRNA, mitochondria, tumor metabolism

## Abstract

**Simple Summary:**

Cellular and mitochondrial metabolism can be dysregulated during tumorigenesis. miR-27a plays a central role in redirecting cell metabolism in colorectal cancer. In this study, we searched for new miR-27a targets that could influence mitochondria and identified FOXJ3 a master regulator of mitochondrial biogenesis. We validated *FOXJ3* as an miR-27a target in an in vitro cell model system that was genetically modified for miR-27a expression and showed that the miR-27a/FOXJ3 axis down-modulates mitochondrial biogenesis and regulates other members of the pathway. The miR-27a/FOXJ3 axis also influences mitochondrial dynamics, superoxide production, respiration capacity, and membrane potential. A mouse xenograft model confirmed that miR-27a downregulates FOXJ3 in vivo and a survey of the TCGA-COADREAD dataset supported the inverse relationship of FOXJ3 with miR-27a and the impact on mitochondrial biogenesis. The miR-27a/FOXJ3 axis is a major actor in regulating mitochondrial homeostasis, and its discovery may contribute to therapeutic strategies aimed at restraining tumor growth by targeting mitochondrial activities.

**Abstract:**

miR-27a plays a driver role in rewiring tumor cell metabolism. We searched for new miR-27a targets that could affect mitochondria and identified FOXJ3, an apical factor of mitochondrial biogenesis. We analyzed FOXJ3 levels in an in vitro cell model system that was genetically modified for miR-27a expression and validated it as an miR-27a target. We showed that the miR-27a/FOXJ3 axis down-modulates mitochondrial biogenesis and other key members of the pathway, implying multiple levels of control. As assessed by specific markers, the miR-27a/FOXJ3 axis also dysregulates mitochondrial dynamics, resulting in fewer, short, and punctate organelles. Consistently, in high miR-27a-/low FOXJ3-expressing cells, mitochondria are functionally characterized by lower superoxide production, respiration capacity, and membrane potential, as evaluated by OCR assays and confocal microscopy. The analysis of a mouse xenograft model confirmed *FOXJ3* as a target and suggested that the miR-27a/FOXJ3 axis affects mitochondrial abundance in vivo. A survey of the TCGA-COADREAD dataset supported the inverse relationship of FOXJ3 with miR-27a and reinforced cellular component organization or biogenesis as the most affected pathway. The miR-27a/FOXJ3 axis acts as a central hub in regulating mitochondrial homeostasis. Its discovery paves the way for new therapeutic strategies aimed at restraining tumor growth by targeting mitochondrial activities.

## 1. Introduction

microRNAs (miRNAs) are short non-coding RNAs of 21–23 nucleotides in length that are able to modulate gene expression at the post-transcriptional level. They recognize specific sequence motifs mostly located within the 3’ untranslated region (3’UTR) of the target mRNAs, leading to either mRNA degradation or impaired translation [1,2]. Each miRNA has multiple targets, and a single mRNA is recognized by numerous miRNAs.

miRNAs regulate a myriad of cellular processes, including proliferation, differentiation, and development [1,2]. Specifically, during tumorigenesis, they stimulate or inhibit proliferation via interactions with oncogenes or tumor-suppressor genes, respectively, ultimately impacting relevant cellular pathways [3]. miRNAs are emerging as pivotal regulators of tumor cell metabolism rewiring, acting on various and different targets including mitochondria [4,5].

Mitochondria are one of the most important organelles within the cells of multicellular organisms, where they operate as the “powerhouse” of metabolism, catalyzing the production of ATP via oxidative phosphorylation (OXPHOS). Their number varies according to the cell type, its energetic requirements, and in response to intrinsic or extrinsic signaling cues, and this number is guaranteed through the balancing of two distinct processes. Mitochondrial biogenesis implies the formation of new organelles from pre-existing or newly synthesized components via the action of several proteins that control the synchronous transcription and translation of nuclear and mitochondrial genes, as well as mitochondrial DNA replication [6,7]. Mitochondrial dynamics implies cycles of the fusion and fission of existing organelles. The fusion allows for the mixing of mitochondrial DNA and proteins, as well as those involved in OXPHOS, between neighboring mitochondria [8,9]. It occurs between damaged and healthy mitochondria and helps to buffer transient stresses or defects within a mitochondrion by diluting toxins and acute damages [10]. The fission enables the mitochondria to divide, facilitates mitochondrial traffic, and is crucial to maintain organelle distribution in the cell and daughter cells in mitosis. It also allows for the segregation and elimination of unhealthy components through mitophagy [11]. Mitochondrial biogenesis and dynamics are finely regulated; alterations of both processes are associated with dysfunctional mitochondria and disease states [6,9,10,12]. These processes also play a role during tumorigenesis with contrasting results, as they may be both positive and negative regulators according to cancer type [13]. In colorectal cancer (CRC), a type of tumor with the highest incidence, mortality, and morbidity rates worldwide [14], the limitation of the bioenergetic activity of mitochondria is associated with tumor progression, and tumors with a low bioenergetic signature have a worse prognosis [15].

In this study, we sought to identify novel miR-27a targets able to influence mitochondrial structure and functions in CRC, as well as to provide mechanistic insights. miR-27a has been shown to regulate fuel preference in post-mitotic muscle cells, influencing fiber-specific regulatory networks and mitochondrial morphology [16]. Muscle physiology and mitochondrial activities are also modulated by miR-27b [17]. We previously reported that miR-27a acts as a driver oncogene in CRC and plays a pivotal role in redirecting cell metabolism [18,19]. Here, we analyzed a list of putative miR-27a targets and selected those exclusively implicated in mitochondrial functions. By using bioinformatic tools and functional analysis, we identified “cellular component organization or biogenesis” as the top pathway, and we focused on Forkhead Box J3 (*FOXJ3*) as the most upstream regulatory gene. FOXJ3 is a member of the Forkhead box (FOX) large family of transcription factors characterized by an evolutionarily conserved winged helix DNA binding domain that recognizes cis-regulatory elements in target gene promoters. To date, fifty mammalian FOX proteins have been identified and classified based on their sequence homology within the winged helix and other functional domains [20]. Most members are involved in cell differentiation during embryonic development and in cell proliferation and apoptosis during adulthood. Accordingly, a number of studies have linked the deregulation of FOX factors with malignant transformation in which they act as tumor suppressors [21]. Importantly, in addition to its roles in cell cycle control [22], cell proliferation, and several cancer types [23,24], FOXJ3 has been found to be an upstream transcriptional activator of mitochondrial biogenesis [17,23]. We verified that *FOXJ3* is an miR-27a target in an in vitro CRC cell model system and validated it by specific target protector oligonucleotides. We also showed that the miR-27a/FOXJ3 axis impairs mitochondrial biogenesis and dynamics, superoxide production, OXPHOS activity, and membrane potential. The analysis of a mouse xenograft model confirmed the inverse relation with FOXJ3, suggesting that the miR-27a/FOXJ3 axis influences mitochondrial abundance in vivo. The same inverse correlation of miR-27a with FOXJ3 was found in CRC patients after investigating the TCGA-COADREAD dataset, and “cellular component organization or biogenesis” was the most down-modulated pathway. Collectively, our data show that miR-27a and FOXJ3 negatively orchestrate overall mitochondrial homeostasis.

## 2. Materials and Methods

### 2.1. Identification of Predicted Targets

The miRWalk tool [25] was used for the analysis of miR-27a-3p target prediction. A combination of genetic pathway databases and manual curations from a literature search was used for the functional analysis of all predicted targets (IMPI, MitoCarta, BioCarta, Reactome, WikiPathways, and Europe PMC publications). Enrichment Pathway Analysis was conducted using the Metascape tool (www.metascape.org) (accessed on 10 January 2020). [26]. The search for miR-27a-3p recognition seeds on target genes was performed using IntaRNA [27] (http://rna.informatik.uni-freiburg.de/IntaRNA/Input.jsp) (accessed on 7 January 2020). Target sequence conservation across species was assessed using EBi alignment tools.

### 2.2. Cell Culture and Target Site Blockers

The human CRC cell lines HCT116 and HT29 were acquired from the American Type Culture Collection (ATCC, Rockville, MD, USA). The miR-27a-overexpressing or -silenced cell clones were obtained and cultured as previously described [19]. miRCURY LNA miRNA Power Target Site Blockers (TSBs) (Qiagen cod. 339194, Hilden, Germany) and a negative control were transfected following the manufacturer’s instructions. The TSB sequences, reported in Appendix A, spanned the recognized seed motifs and extended for 20–23 nucleotides on both sides to obtain a specific protection of the selected mRNA. Moreover, the manufacturer (Qiagen) employs LNA oligonucleotides with a substantially increased affinity for their complementary strand compared to that of traditional DNA or RNA oligonucleotides, resulting in unprecedented sensitivity and specificity.

### 2.3. Gene Expression Profiling, mRNA, Nuclear and Mitochondrial DNA Quantitation

DNA and RNA were extracted using a DNA extraction kit (D3004, Zymo, Irvine, CA, USA) and TRIZOL® Reagent (Invitrogen, Carlsbad, CA, USA), respectively, following the manufacturer’s instructions. DNA and RNA purity and quantity were assessed as previously described [28]. Mitochondrial DNA content was evaluated with a commonly used method based on the mitochondrial to nuclear DNA (mtDNA/nDNA) ratio, in which we quantified the mitochondrial encoded genes *tRNA*^Leu^ and *MT-RNR2* versus nuclear encoded ones, *Claudin1* and *SOX9*, by q-PCR [29]. The sequences of the specific primers for DNA and RNA analysis are reported in Appendix A.

### 2.4. Western Blot Analysis

Protein extracts from cell lines were analyzed as previously reported [19]. The used antibodies are listed in Appendix A. Some blots were cut and probed with different antibodies for different proteins, including β-actin and α-tubulin. In some cases, to examine proteins of similar molecular weight, the PVDF membranes were subjected to a mild stripping protocol, as recommended by Abcam. The obtained bands were densitometrically quantified with Image Lab software (Bio-Rad, Hercules, CA, USA).

### 2.5. Confocal Microscopy Image Acquisition

Cells were stained with a 2.5 µM MitoSOX probe (M36008, Thermo Fisher Scientific, Waltham, MA, USA) or a 200 nM TMRE probe (T669, Thermo Fisher Scientific) for 20 min at 37 °C and examined using a confocal microscope (TCS SP8; Leica, Wetzlar, Germany). Hoechst (62249, Thermo Fisher Scientific) was used to visualize the nuclei, as previously described [30]. 3D reconstruction was assessed using Leica LasX 3D software. For TOM-20 immunofluorescence staining, cells were analyzed according to manufacturer’s instructions. To evaluate mitochondrial fragmentation, approximately 100 mitochondria in three randomly chosen fields were selected, and their size/shape was measured with ImageJ software (https://imagej.net/imaging/particle-analysis) (accessed on 9 January 2020). The percentage of mitochondria smaller than 0.6 μm^2^ was reported as the Circularity Factor.

### 2.6. Seahorse XFe96 Metabolic Assays

We seeded 3 × 10^4^ cells/well in XFe96 cell culture plates, and 24 h later, we replaced the medium with an XF base medium supplemented with 2 mM glutamine, 1 mM sodium pyruvate, and 25 mM glucose. Cells were then incubated in a non-CO_2_ incubator for 1 h at 37 °C to pre-equilibrate the cells before analysis. An XF Mito Stress Test was performed to assay the cells’ ability to exploit mitochondrial oxidative metabolism, according to the manufacturer’s instructions [31]. This analysis was performed via the real-time measurement of extracellular acidification (ECAR) and oxygen consumption rate (OCR) after the injection of a sequence of compounds that interfere with the electron transport chain: oligomycin (1 µM), carbonyl cyanide-4 (trifluoromethoxy) phenylhydrazone (FCCP) (1 µM), and Rotenone/Antimycin A (0.5 µM). Protein quantification was used to normalize the results. The OCR/ECAR ratio was calculated by considering measurements at the basal condition. Maximal respiration was calculated as the average of three measurements performed after FCCP injection minus the average of three measurements performed after Rotenone/Antimycin A injection.

### 2.7. In Vivo Experiments

Western blots were performed on protein extracts of tumors from immunocompromised mice injected with HCT116 or HT29 cells and treated with an miR-27a anti-sense, mimics, or scrambled controls, as previously described [18,19]. Animal experiments, performed in duplicate, were reviewed and approved by the Ethics Commission at Menarini Ricerche according to the guidelines of the European Directive (2010/63/UE). No adverse or toxic effects were observed.

### 2.8. TCGA-COADREAD Data Set Analysis

Data from The Cancer Genome Atlas (TCGA) consortium (https://portal.gdc.cancer.gov/) (accessed on 27 February 2020) were retrieved, and Colon and Rectum Adenocarcinomas (COADREAD) IlluminaHiSeq and/or Illuminaga mRNA and miRNA expression profiles were obtained from patients for which both data were available (N = 548). COADREAD patients’ RNA-Seq was grouped by miR-27a median value (High (H) or Low (L)) and analyzed by applying the Mann–Whitney U test. Then, differential gene expression was evaluated.

### 2.9. Statistics

Statistical analyses were conducted using GraphPad Prism 6 (GraphPad Software Inc., San Diego, CA, USA). Data are reported as mean ± SEM of experiments performed at least in duplicate; for Western blots, and OCR analysis, the mean values from miR-27a_KD or OE cells were compared with their relative scramble cells by applying the *t*-test. The same test was used for the TMRE staining, as we compared the Neg_CTR and TSBs transfected in each couple of cells or between the two Neg_CTRs of each cell couple. The mRNA analysis shown in Appendix A was performed using the ANOVA with Dunnett’s post-test. Statistical significance was considered when *p* ≤ 0.05.

## 3. Results

### 3.1. miR-27a Affects Mitochondrial Biogenesis and Structure/Organization In Silico

To identify new miR-27a target(s) that could affect mitochondrial functionality in CRC, we used the miRWalk target prediction tool with the single miRNA search [25]. miR-27a-3p is the predominant form of miR-27a in CRC and affects mitochondrial metabolism through several factors, as we previously reported [18,19]. We found that 11238 genes were recognized as targets and functionally classified by using several pathway databases and/or manually curating data from the literature (see Section 2: Materials and Methods). Out of these, we selected 1335 predicted targets related to the activity and organization of mitochondria, as several human and cancer development pathologies have been correlated with mitochondrial dysfunction. We then used Metascape to perform the functional enrichment analysis of the predicted targets and to identify the most affected pathways [26]. Figure 1A shows gene sets whose members were significantly overrepresented in the input gene list and reported in the bar graph based on their statistical significance. “cellular component organization or biogenesis” turned out to be the pathway most influenced by miR-27a. Among the genes we initially classified as miR-27a targets and implicated in the abovementioned most recognized pathway, we selected FOXJ3 because the literature suggests that it is the leader factor of mitochondrial biogenesis. Thus far, FOXJ3 has been shown to act as a master regulator of this process within skeletal muscle, cardiomyocytes, and neurons [17,32,33]. We thus investigated its possible involvement in CRC.

### 3.2. FOXJ3 Is a Direct Target of miR-27a

To confirm our in silico prediction, we inspected the complete *FOXJ3* mRNA sequence to identify seed sequences putatively bound by miR-27a. By using IntaRNA, a program that rapidly and accurately predicts interactions between RNA molecules [27], we found two elements. The former was found to be located at position 794–816 within the coding sequence (CDS) and exhibits an atypical seed formed by an octamer followed by additional nucleotides at the 3′ end, known as “3′-supplementary site” reported to improve binding specificity and affinity [2]. The latter was found to be located at position 3247–3266 within the 3’UTR and is a canonical one with a conserved octameric sequence at positions 5–12 from the 5′end of the miRNA. Of note, both seed sequences are conserved across species as distant as *Danio rerio* and humans, indicating they are preserved through evolution, likely due to their function (Figure 1B).

We evaluated FOXJ3 expression in our in vitro CRC cell model system, as previously described [19]. Briefly, we transduced the high-miR-27a-expressing HCT116 cells with a viral vector carrying a short hairpin antisense RNA to generate pools of clones with reduced levels, henceforth named miR-27a_KD cells. In contrast, we transduced the low-miR-27a-expressing HT29 cells with a viral vector carrying a mimic RNA to generate pools of clones with enhanced levels, henceforth named miR-27a_OE cells. As a control, we used pools of clones, named Scr_KD and Scr_OE, obtained by transducing a vector carrying scrambled sequences in HCT116 and HT29, respectively. In Scr_KD cells, FOXJ3 expression was assessed as relatively low by Western blot analysis, while FOXJ3 expression in miR-27a_KD cells significantly increased due to miR-27a silencing. Contrariwise, in Scr_OE cells, FOXJ3 expression was high and diminished in miR-27a_OE cells due to miR-27a upregulation (Figure 1C). The qRT-PCR results of *FOXJ3* mRNA paralleled the results obtained with the proteins, i.e., the miR-27a_KD and Scr_OE cells displayed higher expression than their counterparts, indicating that miR-27a affects *FOXJ3* mRNA stability and translation (Figure 1D).

We validated *FOXJ3* mRNA as a direct target of miR-27a by carrying out experiments with a TSB, an oligonucleotide complementary to a predicted seed sequence on the selected target mRNA. TSBs are commonly used to study an miRNA’s function because they enable the assessment of the biological effects originating from the blockade of its interaction with the selected mRNA target without affecting other genes and pathways controlled by the microRNA [18,34,35,36]. We synthesized two TSBs, (referred to as TSB1 and TSB2), each complementary to the predicted seed regions reported above and extending on both sides for at least 20–23 nucleotides to assure specificity and selectivity of binding and to avoid off-target effects. As a control, we adopted a scrambled oligonucleotide that recognizes neither *FOXJ3* mRNA nor other mRNAs sequences (henceforth defined Neg_CTR). The TSBs, alone or in combination, and the Neg_CTR were transfected in our cell system for 72 h, and protein extracts were analyzed by Western blotting.

A combination of TSB1 and TSB2 (henceforth named TSBs) elicited an increase in the FOXJ3 protein, especially in cells overexpressing miR-27a (Scr_KD and miR-27a_OE cells), to the point of almost reaching the level of miR-27a_KD or Scr_OE cells; in these latter cells, with lower miR-27a expression, the increase was modest (Figure 1E). qRT-PCR analysis revealed that the two TSBs also affected *FOXJ3* mRNA, with an increase of about 3-fold both in miR-27a_KD and Scr_KD cells (Appendix A). A similar increase was observed in the HT29 cells. Independent transfections of TSB1 or TSB2 induced lower increases of both *FOXJ3* mRNA and protein than those obtained with the two TSBs together. No substantial changes were detected with the Neg_CTR (Appendix A).

These results suggest that the two TSBs protect *FOXJ3* mRNA and make it available for translation in miR-27a-overexpressing cells (Scr_KD and miR-27a_OE cells) in order to reduce the protein differences detected in basal conditions. The less noticeable effect on the FOXJ3 protein in miR-27a_KD and Scr_OE cells, i.e., in cells with reduced miR-27a levels, may be due to the fact that the corresponding mRNA is already translated at a sustainable level and is not subjected to further increase.

These experiments definitely validated FOXJ3 as an miR-27a target with impacts on the protein and mRNA stability. The two seed sequences appeared to have low and equivalent efficacy when considered separately but were higher in combination, suggesting a cumulative effect for the better silencing of the gene (details in Figure 1B). From these results, we decided to carry out all subsequent experiments with the two TSBs together (TSBs) to better replicate physiological conditions.

### 3.3. miR-27a Regulates Mitochondrial Biogenesis through FOXJ3

FOXJ3 is a master regulator of mitochondria biogenesis in tissues with high metabolism and energy demands [32,33]. In fact, FOXJ3 modulates peroxisome proliferator-activated receptor coactivator1-a (PGC1-α) through members of the MEF2 family of transcription factors (MEF2A and MEF2C) [37,38]. PGC1-α is a transcriptional co-activator expressed in highly metabolic tissues with a pivotal role in many mitochondrial activities [7,39]. PGC1-α, in turn, stimulates the expression of nuclear respiratory factor 1 and 2 (NRF1 and NRF2, respectively), two transcription factors with which it cooperates to activate many nuclear and mitochondrial genes required for biogenesis and respiratory functions [39,40,41]. Furthermore, PGC1-α binds to and enhances the transcriptional activity of NRF1 and NRF2 on the promoter of mitochondrial transcription factor A (TFAM), which is required for the transcription of mitochondrial genes as well as the synthesis and maintenance of mitochondrial DNA [38,39,41]. Moreover, *MEF2A*, *NRF1*, and *PGC1-α* and their corresponding protein products form a mutually reinforcing self-regulatory and cross-regulatory network that is capable of directing OXPHOS in muscle [37,40,42].

We thus evaluated whether miR-27a, through the modulation of FOXJ3, has any role in the biogenetic process in our CRC cell model system. PGC1-α, NRF1, and TFAM were evenly upregulated in miR-27a_KD cells with respect to their relative Scr_KD controls, paralleling FOXJ3 in Western blot analysis. In contrast, they were equally reduced in miR-27a_OE compared to Scr_OE cells (Figure 2A).

Interestingly, the analysis of the corresponding mRNAs via qRT-PCR showed an inverse correlation with miR-27a, suggesting that, in addition to FOXJ3, miR-27a regulates other factors of the pathway at both the RNA and protein levels (Figure 2B). Indeed, *NRF2* and *MEF2C* have already been reported as validated targets of miR-27a [43,44]. We predict here that *PGC1-α*, *NRF1*, and *TFAM* are targets, as per our own bioinformatic analysis (Appendix A). Moreover, through the same analysis, we recognized multiple copies of the miR-27a seed sequences within the mRNAs of all the identified targets, supporting the idea that miR-27a affects stability and translatability.

We also examined whether restoring FOXJ3 expression by TSBs could influence their expression. PGC1-α, NRF1, and TFAM proteins increased in cells, with the higher content of miR-27a (Scr_KD and miR-27a_OE) abrogating the differences with their counterparts in basal conditions. In the cells with lower miR-27a (miR-27a_KD and Scr_OE), only slight or no variations were detected (Figure 2C). Altogether, these results show that miR-27a negatively modulates FOXJ3 and other members of the biogenetic pathway at the mRNA and protein levels. Since FOXJ3, in turn, is able to rescue the other components, it has to be downregulated in order to miR-27a negatively control the overall mitochondrial biogenetic process.

### 3.4. miR-27a Affects Mitochondrial Mass and Dynamics

As the miR-27a/FOXJ3 axis regulates mitochondrial biogenesis, we examined the mitochondrial content by staining our cells with an antibody recognizing TOM20, a member of the multi-subunit TOM complex (preprotein translocases of the outer mitochondrial membrane) [45]. Confocal microscopy analysis showed a strong staining for TOM20 in miR-27a_KD and Scr_OE cells with respect to the corresponding Scr_KD and miR-27a_OE relative counterparts, suggesting an increase in mitochondrial abundance (Figure 3A). We then analyzed the levels of ATP5A, UQCRC2, SDHB, COX II, and NDUFB8 belonging to the electron transport chain complexes V, III, II, IV, and I, respectively, by Western blot. Overall, they showed higher expression in miR-27a_KD cells than the Scr_KD controls, thus indicating a higher mitochondrial content. The opposite profile was obtained for miR-27a_OE compared to Scr_OE cells (Figure 3B).

We subsequently assessed the amount of mitochondrial DNA by establishing the mitochondrial to nuclear DNA ratio following the qPCR of mitochondrial (*tRNA*^Leu^ and *MT-RNR2*) and nuclear encoded genes (*Claudin*1 and *SOX*9) [29]. In miR-27a_KD cells, the mtDNA/nDNA ratio was significantly higher (about 50%) than that of their Scr_KD controls; on the contrary, in miR-27a_OE cells, the mtDNA/nDNA ratio was lower than that of the Scr_OE cells (Figure 3C).

Finally, we verified whether mitochondrial dynamics was also affected by miR-27a, as this event is strictly interconnected with biogenesis. As shown in Figure 3D, we used Western blot analysis and found that the expression of Mitofusin 1 and 2 (MFN1 and 2) and OPA1, major players in mitochondrial membrane fusion [8,46], was higher in miR-27a_KD than the relative Scr_KD cells. In contrast, their expression in miR-27a_OE was lower than that in Scr_OE cells, in an inverse correlation with miR-27a. The Mitochondrial Fission Factor (MFF) was more expressed in Scr_KD than miR-27a_KD cells. MFF is anchored to the outer membrane and recruits the Dynamin-Related Protein 1 (DRP1) that undergoes oligomerization and phosphorylation, thus triggering mitochondrial fission [47,48]. In line, DRP1 displayed higher phosphorylation at S616 (activating) and lower phosphorylation at S637 (inhibitory) in Scr_KD than miR-27a_KD cells. In HT29 cells, MFF and DRP1 phosphorylation showed the opposite behavior: MFF was more expressed in miR-27a_OE cells, and DRP1 exhibited higher phosphorylation at S616 and lower phosphorylation at S637 than its Scr_OE counterpart, in line with miR-27a levels. All these results indicate that miR-27a negatively modulates mitochondrial content, thus favoring fragmentation.

### 3.5. The miR-27a/FOXJ3 Axis Affects Mitochondrial Superoxide Production, Respiration and Membrane Potential

Lastly, we assessed whether the miR-27a/FOXJ3 axis modulates mitochondrial functions. We first appraised the levels of superoxide, a marker of mitochondrial stress, by staining the cells with the matrix-targeted fluorescent probe MitoSOX™ and analyzing them with confocal microscopy imaging. miR-27a_KD cells displayed a stronger fluorescence signal than Scr_KD cells (Figure 4A, left panels). On the contrary, the fluorescence in miR-27a_OE was weaker than that in Scr_OE cells (Figure 4A, right panels), suggesting that cells with lower miR-27a expression (miR-27a_KD and Scr_OE) have a higher oxidative stress, which is presumably linked to a higher respiratory capacity.

We thus evaluated this activity via the OCR with the Seahorse XFe96 Mito Stress assay. miR-27a_KD cells displayed higher values of both basal and maximal respiration, with a good production of ATP and a moderate respiratory capacity compared to Scr_KD cells (Figure 4B). On the contrary, miR-27a_OE cells had a basal and maximal respiration/ATP production lower than Scr_OE, confirming previous data [19]. The reduced mitochondrial respiration correlated with the lower amount of the electron respiratory chain proteins data reported above (Figure 3B), in line with miR-27a expression. Interestingly, rescuing FOXJ3 induced an OCR increase, in particular of the maximal respiratory capacity, in cells overexpressing miR-27a (Scr_KD and miR-27a_OE) (Figure 4C, left panels). In contrast, in HT29 Scr_OE, with the lowest miR-27a levels, the maximal respiratory capacity showed no significant variations (Figure 4C, right panels). Notably, the maximal respiratory capacity also increased in miR-27a_KD cells, likely due to the fact that parental HCT116 cells display the highest miR-27a levels [18] and, despite the silencing, a consistent residual activity persists [19]. The increase of the maximal respiratory capacity observed upon FOXJ3 recovery was accompanied by a similar trend of the OCR/ECAR ratio, suggesting a more prominent dependency on OXPHOS than on glycolysis under these conditions.

Finally, we evaluated the mitochondrial membrane potential (ΔΨ) by using TMRE (tetramethylrhodamine ethyl ester), a red–orange dye that accumulates in mitochondria. By confocal microscopy imaging, we showed that cells with low levels of miR-27a (miR-27a_KD and Scr_OE cells) transfected with the Neg_CTR accumulated more dye than those overexpressing miR-27a (Scr_KD and miR-27a_OE cells), indicating that they had a higher membrane polarization potential and more functional mitochondria (Figure 5A).

Rescuing FOXJ3 in Scr_KD cells increased mitochondrial activity, induced an elongated morphology, and a tendency to form a network throughout the cytosolic compartment. These effects were less pronounced in miR-27a_KD cells because they already exhibit mitochondrial activity and an established network. HT29 cells showed the opposite behavior: in miR-27a_OE cells, the few punctate mitochondria acquired a tubular shape, increased in number, and formed a network; in Scr_OE cells, the changes in abundance and morphology were limited (Figure 5A).

We also assessed mitochondrial fragmentation, as an index of fission, by measuring the size and shape of the organelles in several representative confocal microscopy fields of the same cells stained with TMRE. The histograms show the percentage of mitochondria displaying size/shape, i.e., a Circularity Factor, smaller than 0.6 μm^2^ as likely having undergone fission. Cells overexpressing miR-27a were found to have a high Circularity Factor, and the rescue of FOXJ3 was found to at least in part restore an elongated shape and the network formation.

Altogether, these results indicate that the miR-27a/FOXJ3 axis negatively modulates overall mitochondrial functionality.

### 3.6. FOXJ3 Is an miR-27a Target in a Mouse Xenograft Model in Vivo

To investigate whether FOXJ3 is a target of miR-27a in vivo, we examined the xenografts obtained by implanting HCT116 and HT29 cell lines in immune-deficient mice and intratumorally injecting them with an miR-27a inhibitor, mimic, or corresponding scrambled control, as previously described [18,19]. The intratumoral injection of miR-27a mimics remarkably increased tumor growth (size and volume), as well as proliferative and metabolic markers. In contrast, miR-27a inhibitors produced the opposite results [18,19]. FOXJ3, NRF1, SDHB, and COX II expression showed the same inverse relation with miR-27a in extracts from scrambled RNA injected tumors (N = 3) evaluated by Western blot analysis (Figure 6A). Opposite results were obtained in extracts from tumors (N = 3) injected with the miR-27a mimic or inhibitor, which was in line with the modified levels of miR-27a. These results demonstrate that miR-27a targets FOXJ3 in a mouse xenograft model and suggest that the miR-27a/FOXJ3 axis also affects key markers involved in mitochondrial abundance in vivo.

### 3.7. The miR-27a/FOXJ3 Axis Orchestrates Mitochondrial Organization in a CRC Dataset 

To further confirm that *FOXJ3* is a direct target of miR-27a in vivo, we investigated the TCGA-COADREAD miR/RNA-Seq dataset and performed differential mRNA abundance analysis to identify additional pathways related to miR-27a. The survey, carried out on a large cohort of CRC patients (N = 548), showed a robust inverse correlation between *FOXJ3* and miR-27a expression, thus confirming our prediction analysis and the results obtained in the cell lines reported above (Figure 6B). We then used the Metascape tool [26] in the ontology enrichment analysis to identify the pathways influenced by the DE genes. We stratified the patients on the basis of miR-27a and *FOXJ3* median expression and only selected those with opposite values (out of 548, 158 patients exhibited miR-27a-high/FOXJ3-low expression and 153 exhibited miR-27a-low/FOXJ3-high expression). We found that 2055 genes were differentially expressed between these two groups (see Materials and Methods), and 21 out of the top 22 enriched pathways were the same as those obtained from our prediction analysis in silico (Appendix A). These results strengthen the power of the two independent approaches and support the role that the miR-27a/FOXJ3 axis plays in regulating these biological processes. More importantly, when we only selected mitochondrial genes (217) among the total DEGs, “cellular component organization or biogenesis” was found to be the most enriched pathway in the COADREAD dataset, further validating our initial choice (Figure 6C). Altogether, these results demonstrate that the miR-27a/FOXJ3 axis acts as a master regulator of mitochondrial homeostasis in CRC, both in vitro and in vivo.

## 4. Discussion

In this study, we identified *FOXJ3* as a novel target of miR-27a and showed that the pathway “cellular component organization or biogenesis” is primarily affected by the miR-27a/FOXJ3 axis in CRC, with the down-modulation of mitochondrial biogenesis and the upregulation of mitochondrial fission and dysfunctions among the top processes.

We initially predicted *FOXJ3* as an miR-27a target by surveying available algorithms and subsequently validated it in our in vitro CRC cell model system. miR-27a regulates FOXJ3 at both the mRNA and protein levels, suggesting a stringent control mediated by the two seed sequences present in the transcript that likely act in a cumulative manner. The affinity and the energetic and binding parameters towards miR-27a were found to be similar. FOXJ3 is a member of the large FORKHEAD family of transcription factors and has been reported so far to mainly stimulate mitochondria biogenesis in muscle, neuronal, and heart tissues, actively driving differentiation. We have provided evidence that miR-27a targets not only FOXJ3 but also other factors of the cascade at both the mRNA and protein levels, likely through the multiple recognition sequences on the corresponding mRNAs, so the miR-27a/FOXJ3 axis downregulates overall mitochondrial biogenesis (Figure 6D). Interestingly, the rescue of FOXJ3 is associated with the recovery of other proteins of the pathway. FOXJ3 is the most upstream factor and regulates *PGC1-α* and *MEF2C* transcription [37,39]. PGC1-α, in turn as a transcriptional coactivator, modulates the expression of NRF1 and NRF2 that, together with PGC1-α, stimulate TFAM [39,40,41]. Finally, FOXJ3 directly regulates TFAM, as reported in the ENCODE repository [49]. These results support FOXJ3 as a driver gene in mitochondrial biogenesis in our cell system and highlight the relevance of its down-modulation by miR-27a to tone down the overall biosynthetic process. These results also suggest that multiple and complex levels of regulation exist and that additional controls to finely tune the process cannot be ruled out. Mitochondrial dynamics, a process tightly interconnected with biogenesis, is also modulated by the miR-27a/FOXJ3 axis through several factors that inhibit fusion and favor fission events, exerting a negative control on mitochondrial abundance. Notably, in muscle cells, MFN1 and 2 (the two mitochondrial fusion players) are under the transcriptional control of PGC1-α that is downregulated by miR-27a, at least in part explaining the negative impact on the process [10]. Consistently, mitochondria with reduced size and shape are predominant in cells with high miR-27a/low FOXJ3 levels, suggesting that they more likely undergo fission. FOXJ3 rescuing reestablishes an elongated, tubular shape and the formation of a network. Accordingly, in the same cells, functional parameters such as superoxide content, OXPHOS, and mitochondrial membrane potential are also downregulated.

The miR-27a/FOXJ3 axis thus negatively impacts overall mitochondrial structure/function. The recovery of FOXJ3 restores these characteristics and supports the idea that this gene has to be stringently down-modulated so that miR-27a can achieve its final effect.

The altered balance between biosynthetic and degradative processes with reduced mitochondria biogenesis has been linked to diverse pathologic conditions and cancer [50,51]. In this latter case, the major molecular events underlying many different tumor types are oncogenic *RAS* mutations, higher ERK activity, and DRP1-S616 phosphorylation [10]. A similar mutational landscape is present in HCT116 cells and correlates with high miR-27a and low FOXJ3 expression levels. HT29 cells have a different mutational profile from HCT116 cells with no *RAS* mutations, low miR-27a expression, and a relatively high expression of FOXJ3. We have shown here that manipulating the miR-27a/FOXJ3 axis can enable the modification of most of the mitochondrial characteristics, which is consistent with data showing that reduced mitochondrial abundance and functionality are associated with a higher proliferation potential [10,13]. In addition, excessive fission and reduced mitochondrial size/shape appear to be a cellular adaptation to avoid apoptosis and enhance proliferation and cell survival [52], as shown here in the high miR-27a-/low FOXJ3-expressing cells.

miR-27a was found to target FOXJ3 in a mouse xenograft model, suggesting a role for the miR-27a/FOXJ3 axis in the control of mitochondrial abundance in vivo. Finally, the analysis of the TCGA-COADREAD dataset corroborated the inverse relationship between miR-27a and *FOXJ3* and underlined that, in a large cohort of patients, the primary pathways influenced by the miR-27a/FOXJ3 axis overlap those identified in the list of predicted miR-27a target genes. The same outcome from two independent approaches strongly supports the power of the methods used and the relevance of the results obtained in this study.

## 5. Conclusions

The miR-27a/FOXJ3 axis is a key player in downregulating mitochondrial homeostasis in a stringent and coordinated manner. These activities are part of the more general action of miR-27a as master modulator of CRC metabolism rewiring to support increased biosynthesis of macromolecules for tumor progression [19]. Many of these activities, especially those governing mitochondrial structure/function, are mediated through FOXJ3, linking the mitochondrial to the overall cell metabolism. The disclosure that the miR-27a/FOXJ3 axis is pivotal in modulating mitochondrial functionality adds to our understanding on the molecular events underlying tumorigenesis and may pave the way for further studies aimed at restraining tumor growth by stimulating mitochondrial activities.

## Figures and Tables

**Figure 1 cancers-13-04994-f001:**
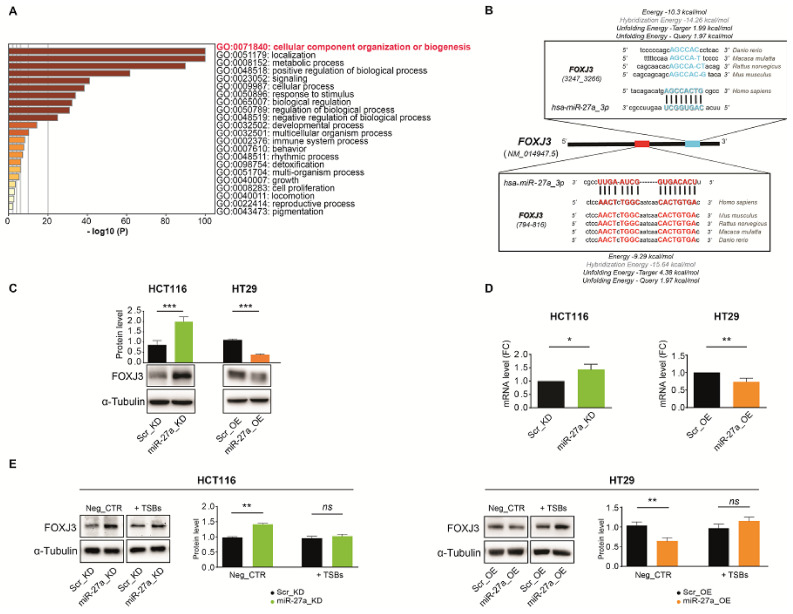
Recognition of the pathways modulated by FOXJ3, identification of the seed sequences for miR-27a on *FOXJ3* mRNA, and expression in a CRC cell model system in vitro. (**A**) The bar graph illustrates the top Gene Ontology biological processes, identified via Metascape, using a discrete color scale to represent statistical significance (a deeper color indicates a smaller *p*-value). (**B**) *FOXJ3* mRNA contains two seed sequences for miR-27a-3p. The characteristics of the binding were calculated by using the IntaRNA algorithm. The miR-27a recognition sequences are highly conserved across species. (**C**) FOXJ3 expression evaluated as protein by Western blot and (**D**) as RNA by qRT-PCR in miR-27a_KD and miR-27a_OE compared to their corresponding Scr_KD and Scr_OE controls under basal conditions. (**E**) FOXJ3 levels in the same cell lines as in (**C**) transfected with the two TSBs or the Neg_CTR. The results shown in panels (**C**) and (**E**) are representative of at least two performed experiments, normalized to the mean ± SEM, and expressed as protein levels with respect to α-tubulin as a loading control. Statistical significance was considered when * *p* ≤ 0.05, ** *p* ≤ 0.01, or *** *p* ≤ 0.001 (*t*-test); *ns* = not significant. Appendix A: Original Western blots.

**Figure 2 cancers-13-04994-f002:**
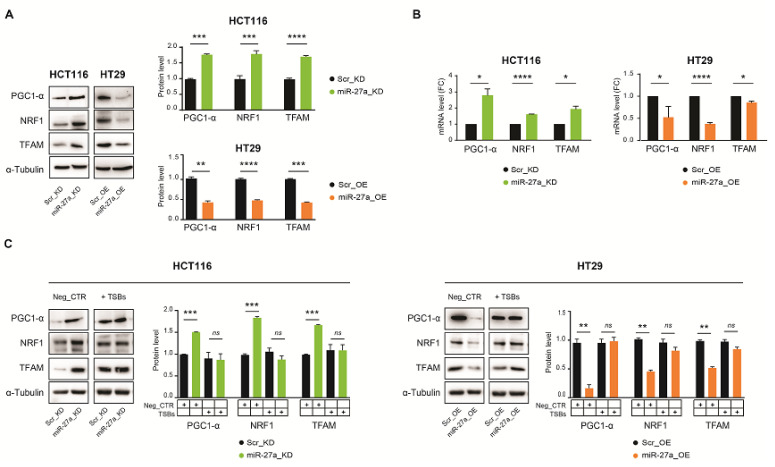
The miR-27a/FOXJ3 axis affects mitochondrial biogenesis. (**A**) Immunoblot and (**B**) qRT-PCR analysis of PGC1-α, NRF1, and TFAM as mitochondrial biogenesis markers in miR-27a_KD and miR-27a_OE cells with respect to their relative Scr_KD and Scr_OE controls in basal conditions. (**C**) Assessment of the same markers as in (**A**) after the transfection of the TSBs or the Neg_CTR. The results shown in (**A**) and (**C**) are representative of at least two performed experiments, normalized to the mean ± SEM, and expressed as protein levels with respect to α-tubulin as a loading control. The α-tubulin shown is from a representative experiment. Statistical significance was considered when * *p* ≤ 0.05, ** *p* ≤ 0.01, *** *p* ≤ 0.001, or **** *p* ≤ 0.0001 (*t*-test); *ns*= not significant. Appendix A: Original Western blots.

**Figure 3 cancers-13-04994-f003:**
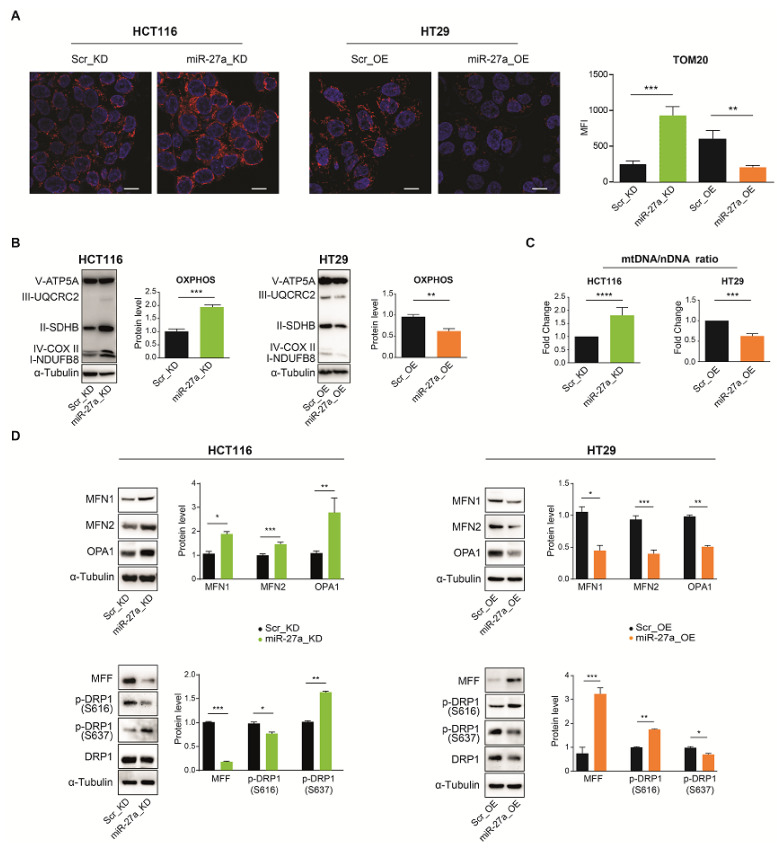
The miR-27a/FOXJ3 axis affects mitochondrial abundance in the CRC cell model system in vitro. (**A**) Immunofluorescence analysis of TOM20 in miR-27a_KD and miR-27a_OE and the corresponding Scr_KD and Scr_OE controls (magnification: 63×; scale bar: 5 μm). The panel on the right illustrates the relative quantification as mean fluorescence intensity (MFI). (**B**) Representative immunoblots in the same cells as in (**A**) performed for the proteins ATP5A, UQCRC2, SDHB, COX II, and NDUFB8 belonging to the respiratory chain complexes V, III, II, IV, and I, respectively. The histograms report the overall OXPHOS protein quantification with respect to α-tubulin as a loading control. (**C**) Evaluation of the mitochondrial DNA/nuclear DNA ratio (mtDNA/nDNA ratio) by qPCR. Data are relative to the mean ± SEM of experiments performed at least in triplicate and expressed as fold-change with respect to the corresponding control cells. (**D**) Analysis of mitochondrial dynamics: Western blot analysis of MFN1, MFN2, and OPA1 as representative of fusion markers (upper panel) and MFF, p-DRP1 (S616), p-DRP1 (S637), and DRP1 as representative of fission markers (lower panel) in the same cells as in (**A**). Panels (**B**) and (**D**) show representative results of at least two performed experiments, normalized to the mean ± SEM, and expressed as protein levels with respect to α-tubulin as a loading control. The α-tubulin shown is from a representative experiment. Statistical significance was considered when * *p* ≤ 0.05, ** *p* ≤ 0.01, *** *p* ≤ 0.001, or **** *p* ≤ 0.0001 (*t*-test). Appendix A: Original Western blots.

**Figure 4 cancers-13-04994-f004:**
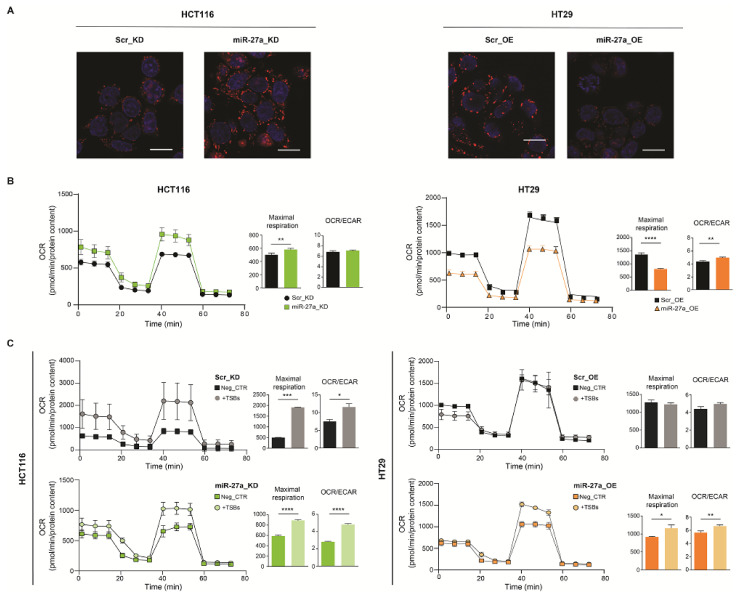
Mitochondrial superoxide production and OXPHOS activity are negatively regulated by miR-27a via FOXJ3. (**A**) Representative confocal microscopy images of the MitoSOX staining (red) of miR-27a_KD and miR-27a_OE, as well as relative Scr_KD and Scr_OE controls. Hoechst was used to stain nuclei (blue) (magnification: 63×; scale bar: 5 μm). (**B**) Oxygen consumption rate (OCR) assay performed in the same cells as in (**A**) in basal conditions or (**C**) following transfection with the TSBs or the Neg_CTR. The histograms report the rates of maximal respiratory capacity and the OCR/ECAR ratios quantified upon the normalization of OCR to O.D. protein levels. Statistical significance was considered when * *p* ≤ 0.05, ** *p* ≤ 0.01, *** *p* ≤ 0.001, or **** *p* ≤ 0.0001 (*t*-test).

**Figure 5 cancers-13-04994-f005:**
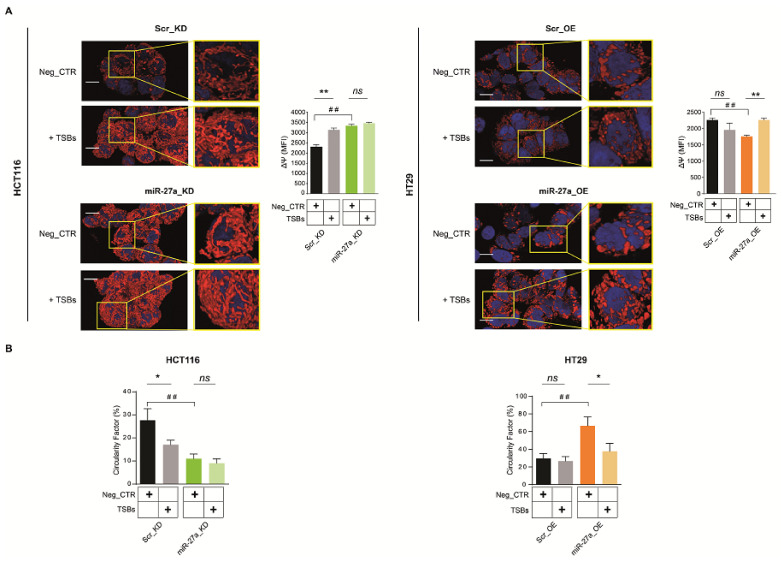
The miR-27a/FOXJ3 axis downregulates mitochondrial outer membrane potential. (**A**) Confocal fluorescence microscopy images of miR-27a_KD and miR-27a_OE cells and their relative Scr_KD and Scr_OE controls stained with TMRE (red) after the transfection of the TSBs or the Neg_CTR. Hoechst was used to stain nuclei (blue) (magnification: 63×; scale bar: 2 μm). The histograms on the right of both panels illustrate the values of the mitochondrial membrane potential calculated as the mean fluorescence intensity (MFI) with ImageJ software, as described in Materials and Methods. (**B**) The histograms show the percentage of mitochondria smaller than 0.6 μm^2^ reported as Circularity Factor for Neg_CTR and TSBs transfected in each couple of cells or between the two Neg_CTRs of each cell couple. Statistical significance was considered when * or # *p* ≤ 0.05, ** or ## *p* ≤ 0.01, (*t*-test) where the * refers to the comparison between Neg_CTR and TSBs transfected for each couple of cells and # refers to the comparison between the two Neg_CTRs of each cell couple.

**Figure 6 cancers-13-04994-f006:**
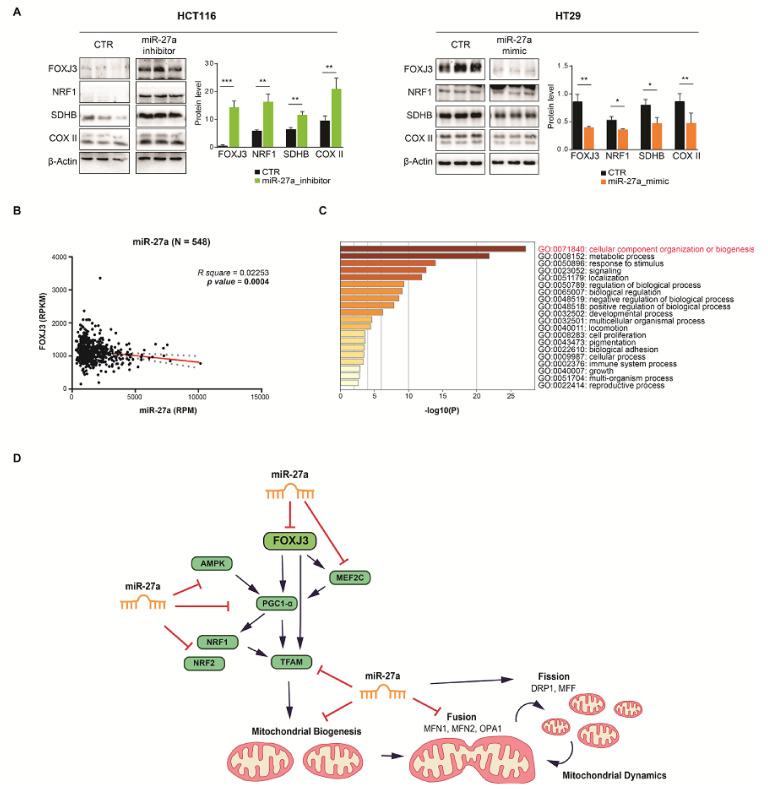
FOXJ3 is a target of miR-27a in a mouse xenograft model in vivo and in a CRC dataset. (**A**) Representative immunoblot for FOXJ3, NRF1, SDHB, and COX II on extracts from xenografts of HCT116 and HT29 cells in immune-compromised mice that were intratumorally injected with an miR-27a inhibitor or mimic, respectively, or scrambled controls. The data shown are the mean of experiments ± SEM performed on extracts from N = 3 mice per each type, expressed as protein levels with respect to β-actin as a loading control. Statistical significance was considered when * *p* ≤ 0.05, ** *p* ≤ 0.01, or *** *p* ≤ 0.001 (*t*-test). (**B**) Correlation of *FOXJ3* with miR-27a expression in TCGA-COADREAD patients; N = 548 and *p* = 0.0004. (**C**) The bar graph illustrates the top Gene Ontology biological processes, identified via Metascape, using a discrete color scale to represent statistical significance (a deeper color indicates a smaller *p*-value). (**D**) This drawing depicts the key factors involved in mitochondrial biogenesis and dynamics, their reciprocal modulation, and their overall regulation by miR-27a. Appendix A: Original Western blots.

## Data Availability

The data presented in this study are available on request from the corresponding author. The data are not publicly available because investigation is underway on novel miR-27a mitochondrial targets.

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
