# Peer review of "The miR-27a/FOXJ3 Axis Dysregulates Mitochondrial Homeostasis in Colorectal Cancer Cells"

_cancers, 2021, doi:10.3390/cancers13194994_

Round 1

Reviewer 1 Report

The manuscript titled: “The miR-27a/FOXJ3 axis dysregulates mitochondrial homeostasis in colorectal cancer cells” by Giovannina Barisciano et al describes a wide study aimed at answering the question whether the FOXJ3 transcription factor could be a target of miR-27a,  which has been known to play an important role in changing metabolism in colorectal cancer (CRC). The FOXJ3 has been found to function in cell cycle control and proliferation, and  to be  upstream activator of mitochondrial biogenesis. The aim of this work is justified, since solving the regulatory pathways in cancer metabolism, including the CRC, is important for understanding the process of tumorigenesis and finding new therapeutic targets.

The Authors applied in silico methods to predict that miR-27a  targets FOXJ3 and then proved this using CRC cells. By various methods, they showed in the CRC in vitro model that miR-27a/FOXJ3 axis down-modulates mitochondrial biogenesis and regulates other members of the pathway. They also showed that the miR-27a/FOXJ3 axis also influences mitochondrial dynamics, superoxide production, respiration capacity and membrane potential.  Next, they used the mouse xenograft CRC model and confirmed that miR-27a targets FOXJ3  in vivo.

These findings are new and well documented. However, I have some comments.

Major comment:

The section 3.6, describing results obtained in vivo. The Authors stated that  FOXJ3 is a miR-27a target in a mouse model in vivo and this statement is supported by the experimental evidence. However, further on, they stated that “the miR-27a/FOXJ3 axis affects key markers involved in mitochondrial abundance”. With the latter statement I can not agree, since it has not been proven. In the mouse model they did show that miR-27a down-regulates FOXJ3, NRF1, SDHB and COX II but did not show that FOXJ3 affects the downstream elements. They only did this in vitro, by demonstrating that restoring FOXJ3 expression had an effect on the downstream proteins (PGC1-α, NRF!, TFAM). Thus, the Authors should change their conclusion. They should write (in all sections) that their results suggest that the miR-27a/FOXJ3 axis affects key markers involved in mitochondrial abundance in vivo.

Minor comment

In the Materials & Methods section the method of assaying the mitochondrial to nuclear DNA ratio should be included.

Author Response

We would like to express our gratitude to Reviewer #1 for the criticisms raised and suggestions to improve the quality of the paper. We have amended the manuscript taking into considerations the criticisms raised and the point-by-point response to His/Her questions is appended as pdf file.

Reviewer 2 Report

In this article, the authors have shown FOXJ3 as a miR27a target and the role of the miR27a/FOXJ3 axis in the regulation of mitochondrial biogenesis. Overall, it is a nice study with well-designed experiments.

 A few comments to the authors-

  1. It is not clear how authors have determined FOXJ3 as the leading factor of the "cellular component organization or biogenesis pathway".
  2. As miR27b is also known to target FOXJ3, why authors have only focussed on miR27a? Did the authors analyze the role of miR27b in the regulation of mitochondrial biogenesis in CRC cells?
  3. Did the authors analyze the effects of miR27a knockdown and overexpression on tumor growth?

Author Response

We would like to express our gratitude to Reviewer #2 for the criticisms raised and suggestions to improve the quality of the paper. We have amended the manuscript taking into considerations the criticisms raised and the point-by-point response to His/Her questions is appended as pdf file.

Reviewer 3 Report

The study of Dr Barisciano et al aimed at identifying miR-27a targets that might affect mitochondria activities. They identified FOXJ3, and evidenced that miR-27a/ FOXJ3 axis impaired mitochondrial biogenesis and function.

This study involved an in silico analysis, and two experimental models, the HCT116 colon cancer cell lines and their miR-27a knock-down derivatives, and the HT-29 colon cancer parental cells -that poorly expressed miR-27a- and derivatives overexpressing this miRNA. The experimental design is accurate and the data clearly exposed.

Comments

The Figures are too small and the text unreadable. Please, enlarge the Figures and increase the font size.

In silico analyses suggest that beside FOXJ3, miR-27a targets other effectors of FOXJ3, including PGC1-a, NRF1 and TFAM (Supplementary Fig2B). Nevertheless, since TSB transfection and restoration of FOXJ3 expression curtails the down-regulation of PGC1-a, NRF1 and TFAM (Figure 2C), the corresponding transcripts do not seem to be direct target of miR-27a.

Chapter 3.6. The title is a shortcut and is a little bit misleading, since the chapter concerns xenografts of human colon cancer cells in nude mice. It would be interesting to know if miR-27a mimic and inhibitor injections impact tumor grouth.

In silico analysis of the TCGA-COADREAD database revealed an inverse relationship of FOXJ3 and miR-27a mRNA levels in colorectal cancers. The Authors could have also reported the relationship with some FOXJ3 downstream targets.

In Figure 5, four groups are compared. The statistical analysis should have been performed using ANOVA or a Kruskal- Wallis test followed with post-hoc tests. There is only mention of t-test in the Material & Methods section. The same comment applies for supplementary Figure 1A.

Colorectal cancers are now subdivided in 4 subgroups, based on consensus molecular signature (CMS1-4). It could be interesting to determine if FOXJ3 downregulation fits with cms3 (metabolism signature) frequently associated with KRAS activating mutation.

Author Response

We would like to express our gratitude to Reviewer #3 for the criticisms raised and suggestions to improve the quality of the paper. We have amended the manuscript taking into considerations the criticisms raised and the point-by-point response to His/Her questions is appended as pdf file.
